# Changes in Cat Facial Morphology Are Related to Interaction with Humans

**DOI:** 10.3390/ani12243493

**Published:** 2022-12-10

**Authors:** Madoka Hattori, Atsuko Saito, Miho Nagasawa, Takefumi Kikusui, Shinya Yamamoto

**Affiliations:** 1Wildlife Research Center, Kyoto University, 2-24 Tanaka-Sekiden-cho, Sakyo-ku, Kyoto 606-8203, Japan; 2Department of Animal Science and Biotechnology, Azabu University, 1-17-71 Fuchinobe, Chuo-ku, Sagamihara-shi 252-5201, Japan; 3Department of Psychology, Faculty of Human Sciences, Sophia University, 7-1 Kioicho, Chiyoda-ku, Tokyo 102-8554, Japan; 4Institute for Advanced Study, Kyoto University, Yoshida Ushinomiya-cho, Sakyo-ku, Kyoto 606-8501, Japan

**Keywords:** cats, face, morphology, domestication, feralization, transgenerational change, domestication syndrome

## Abstract

**Simple Summary:**

Cats (*Felis silvestris catus*) live with humans as domesticated animals. However, it is not clear how they have evolved from African wildcats (*Felis silvestris lybica*) through domestication. In this study, we compared the facial morphology of cats and wildcats to determine how the facial morphology of cats has changed in relation to their interactions with humans. Cats kept by humans had a smaller nose length and a lower eye angle compared with wildcats and feral cats, and these changes were found to influence cuteness ratings. The facial morphology of feral cats not living with humans was not significantly different from that of wildcats, suggesting that this change is due to the process of feralization.

**Abstract:**

We aimed to clarify the changes in facial morphology of cats in relation to their interactions with humans. In Study 1, we compared the facial morphology of cats (feral mixed breed, owned domestic mixed breed, and owned domestic purebreds) with that of African wildcats. After collecting 3295 photos, we found that owned domestic cats’ noses were significantly shorter than those of African wildcats and feral mixed breed, and there were no significant differences between the latter two. The eye angles were significantly more gradual in owned domestic purebreds than in the other groups. In Study 2, we examined the correlation between facial morphology and years with the owner, and found that the former is not affected by the latter. This suggests that changes in facial morphology are possibly transgenerational changes. The difference in facial morphology between wildcats and owned cats might be caused by domestication, and that between feral cats and owned cats might be due to feralization. In Study 3, we investigated whether cats’ facial features affect cuteness ratings. We asked human participants to evaluate the cuteness of cats’ face images and found that faces with shorter nose lengths were considered cuter. This suggests that owned domestic cats’ facial morphology is preferred by humans.

## 1. Introduction

Humans perceive a variety of information from others’ faces [1]. One of the universal groups of facial features across species is baby schema: large head, big eyes, small nose and mouth, high and protruding forehead, chubby cheeks, short and thick limbs, and plump body shape [2]. Baby schema is known to elicit caretaking behaviors and positive affect from humans [3,4,5,6]. For example, heightened baby schema features of infant faces are rated as cuter and elicit stronger motivation for caretaking compared with unmanipulated faces [7]. Viewing infant faces with heightened baby schema activates the nucleus accumbens, which consists of the brain’s reward system [8]. Furthermore, baby schema cuteness ratings are found across species. Comparing species that require relatively low parental care with those requiring moderately high parental care, humans perceive non-mammals (birds, reptiles) requiring high care as cute and have a desire to hold or pet them [9].

Facial morphology is important for a companion animal in eliciting caretaking behaviors from humans. However, it is not yet clear whether face shape actually elicits caretaking behaviors. Baby schema features are also found in companion and domestic animals, especially in domestic dogs (*Canis lupus familiaris*), which are known to have neotenous faces [10]. The domestication of dogs changes their facial morphology and affects human caretaking behavior [11,12]. Contrastingly, adult humans tend to rate human infant faces as cute, but not animal infant faces [13]. Human infant faces have also been found to activate the premotor cortex and the reward system, unlike animal infant faces [14]. In cats, infant facial features are rated as cuter when emphasized [15], and children were found to gaze for longer at these features [16]; however, another study found that the length of time cats were kept in shelters before being adopted did not vary with their face’s cuteness rating [17].

In this study, we focused on the facial morphology of cats (*Felis silvestris catus*), which is one of the main species of companion animals. The advantages of focusing on the facial morphology of cats are as follows. The ecology of cats differs from that of many other domesticated species, with ancestral African wildcats (*Felis silvestris lybica*) possessing barriers to domestication such as independence, territoriality, and obligate carnivory [18]. Although cats started living with humans around 10,000 years ago [19], they are still in the process of domestication, as humans do not have complete control over their breeding and predation. Moreover, cats are still considered to have a wild aspect [20]. For these reasons, research on cats may help to clarify the ongoing process of domestication.

Although there is evidence of variation in the facial morphology of cats in terms of skull and brain size, the results are inconsistent. Studies have shown that domestic cats have a 23% smaller brain size than European wildcats [21] and a 27.6% smaller brain value relative to body weight than African wildcats [22]. However, it has also been reported that the skull size of domestic cats is indistinguishable from that of African wildcats [23]; this finding has been contradicted by a recent study that found that the skull size of domestic cats is smaller than that of European wildcats (*Felis silvestris*) and African wildcats (*Felis silvestris lybica*), but there is no difference in palate [24]. In addition, it is not known whether this domestication-induced change in brain and skull size corresponds to the baby schema that is thought to elicit caretaking behaviors in humans.

If human interaction indeed influences neotenous facial morphology as well as brain and skull size, what process brought about this change? Changes in facial morphology may include “transgenerational changes” and “individual developments.” On the one hand, changes in domestic animal traits due to artificial selection, or “transgenerational changes,” are known as domestication syndromes [25]. Morphological changes include neotenous head shape, white pigmentation, and drooping ears. In fox domestication experiments, these phenotypes became apparent after around ten generations, when less aggressive individuals were selected to breed [10]. Primates such as humans and bonobos show a tendency toward domestication syndrome [26] and similar phenotypic changes such as skull reduction [22] and white sclerites [27].

On the other hand, “individual developments,” that is, ontogenetic changes, can be found, for example, in the changing of human facial features with age [28]. Orangutan males change their facial morphology, known as the fringe, due to changes in social status [29]. Changes in facial morphology may also be associated with changes in behaviors. For example, an association between dominance and face size has been found in primates such as macaque monkeys and bonobos [30,31]. In humans, it is known that aggression is associated with male facial morphology [32] and that habitual expressions of emotions can change facial features [33].

The aim of this study was to investigate whether human interaction influences changes in the facial morphology of cats. We conducted three studies. In Study 1, we collected facial photographs of groups of cats with different degrees of interaction with humans (African wildcats, feral mixed breed, owned domestic mixed breed, and owned domestic purebreds) and examined differences in their facial morphology. To investigate whether facial changes, if any, are due to ontogenetic changes (or otherwise transgenerational ones), in Study 2, we examined the effect of years with the owner and behaviors on facial morphology. In Study 3, we investigated whether the facial morphology features examined in Study 1 influenced cuteness evaluation by humans.

## 2. Materials and Methods

### 2.1. Study 1

To investigate the association between the degree of human interaction and facial morphology, facial photographs of the ancestral species of cats, African wildcats (*Felis silvestris lybica*) and cats (*Felis silvestris catus*) were collected with their faces and ears facing forward and their mouths closed. Fifty photographs of African wildcats were collected from pictorial books and the Internet. Since cats have diverse ways of interacting with the environment and humans, we divided them into three groups (feral mixed breed, which live independently of humans and human food; owned domestic mixed breed; and owned domestic purebreds), as shown in Figure 1. A total of 150 photos of feral mixed-breed cats were provided by the government and conservation groups from Japanese islands (in Amami-Oshima Island, Kagoshima (129°22′ E, 28°19′ N), Ogasawara Islands, Tokyo (142°11′ E, 26°05′ N), and Mikura Island, Tokyo (139° 36’ E, 33°52’ N) in Japan) and collected from the Internet. For owned domestic mixed breed, we collected 661 photos posted by Japanese cat owners between 27 August and 30 September 2019 by soliciting photos with the Instagram hashtag #nekokao2019. We categorized domestic purebreds into 60 breeds based on the CFA (The Cat Fanciers’ Association, http://www.cfa.org/ accessed on 8 October 2019.) and TICA (The International Cat Association, http://www.tica.org/ accessed on 8 October 2019.). A total of 2434 facial photographs of 60 breeds were collected from Instagram and the Internet (Appendix A).

Based on the baby schema measures [7,16], we mapped eyes and midlines. As cats open and close their eyelids very differently depending on the brightness, the size of the eyes cannot be accurately measured from photographs. Therefore, we measured nose length as a possible indicator of baby schema [2]. Using the measuring tool in Adobe Photoshop 21.2 2, the distance between the eyes (A) and the length from the center line of the eye to the bottom of the nose (B) were measured; hereafter, we refer to “nose length” as nose length adjusted by the cat’s facial size (practically adjusted by the distance between the two eyes), that is, B/A in Figure 2. We also measured the tilt angles of the left and right eyes, which are considered to influence facial impressions in humans [34], and recorded the average value of the eye angles, that is, (C+D)/2 in Figure 2.

Owing to differences in sample size among the four groups of African wildcats, feral mixed breed, owned domestic mixed breed, and owned domestic purebreds, the first three groups were each selected by simple random sampling. Thus, African wildcats (*n* = 50), feral mixed breed (*n* = 20 from the three islands giving a total of *n* = 60), and owned domestic mixed breed (*n* = 60) were selected by simple random sampling; for owned domestic purebreds, mean values for each breed (out of 60 breeds) were used. We used R version 3.6.0 for statistical analysis and ran an analysis of variance (ANOVA), followed by Tukey’s multiple comparison test.

### 2.2. Study 2

We investigated how the number of years the cat lived with the owner at the time of the survey (hereafter, “years with the owner”) is related to facial morphology in owned domestic mixed breed. We used the Feline Behavioral Assessment and Research Questionnaire (Fe-BARQ) [35], a behavioral assessment (Appendix A), and conducted a questionnaire survey of the owners of the 661 cats included in Study 1. Using the data from the 428 cat owners who responded, we examined the association with the facial morphology data measured in Study 1. We ran a generalized linear model (GLM), using a glmer function with Poisson distribution in the lme4 package version 1.1.10 with R version 3.6.0. Explanatory variables were items related to sex, years with the owner, and Fe-BARQ behaviors toward humans (sociability with people, stranger-directed aggression (Appendix A)), while response variables were nose length or eye angles.

### 2.3. Study 3

We conducted an online questionnaire to investigate whether a cat’s neotenous facial morphology is rated as cute by humans. The survey was conducted in Japanese using the questionnaire platform Questant (https://questant.jp/ accessed on 7 October 2019). A total of 355 participants (87 male, 265 female, and 3 others who did not respond) aged 18–72 years (mean: 35.25 ± SD 12.10, median: 37) were recruited individually within the authors’ university classes and online (Appendix A). The survey was conducted between 7 September and 8 October 2021.

We used the same photographs of African wildcats, *n* = 50; feral mixed breed, *n* = 60; owned domestic mixed breed, *n* = 60; and owned domestic purebreds, *n* = 60 as used in Study 1 as stimuli. Photographs of the four groups were processed with the morphing application Average Face PRO (https://apps.apple.com/jp/app/id560519978 accessed on 8 September 2021) to create an average face for each group. Then, to eliminate the influence of differences in pupil size, the processed facial photographs of the four groups were combined into a single photograph using Average Face PRO. The eyes in that photograph were then combined with the eyes in the photographs obtained for each of the four groups using Adobe Photoshop 21.2 2. The images were also processed in black and white to eliminate the influence of coat color.

First, we prepared average faces for each of the four types of cats. We compared all six possible combinations among the four average faces. Next, to investigate the effects of nose length and eye angles, we created an image of an owned domestic purebred cat face with either a longer nose length or larger eye angle. Then, we compared this image with the original average face of owned domestic purebred cats. Thereafter, we created an African wildcat face with either a shorter nose length or smaller eye angle, and compared it with the original average face of African wildcats. Due to copyright issues, only the image reflecting the average face of owned domestic mixed-breed cats is included here (Figure 3).

Statistical analysis was first performed by calculating David’s score [36] from a list of won-and-lost records to determine the rated cuteness rank. Next, in order to examine the effects of manipulating the nose length and eye angles, we conducted a two-tailed binomial test (baseline 0.5) using R version 3.6.0 with Bonferroni correction.

## 3. Results

### 3.1. Study 1

We measured nose length and eye angles for the four groups of cats (Figure 4). Nose length adjusted by face width (i.e., B/A in Figure 2) had the following means: 1.34 (*SD* = 0.11) for African wildcats, 1.32 (*SD* = 0.17) for feral mixed breed, 1.23 (*SD* = 0.14) for owned domestic mixed breed, and 1.14 (*SD* = 0.21) for owned domestic purebreds. The eye angles had the following means: 25.61 (*SD* = 3.06) for African wildcats, 25.10 (*SD* = 3.57) for feral mixed breed, 25.31 (*SD* = 3.67) for owned domestic mixed breed, and 22.77 (*SD* = 2.36) for owned domestic purebreds.

The ANOVA detected significant differences between groups (*F*(3226) = 18.39, *p* < 0.01, *η*^2^ = 0.20) for nose length. Post hoc analyses showed that owned domestic purebreds had a significantly smaller nose length than African wildcats (*p* < 0.01, *d* = 1.2), feral mixed breed (*p* < 0.01, *d* = 0.96), and owned domestic mixed breed (*p* = 0.02, *d* = 0.50). Owned domestic mixed breed had a significantly smaller nose length than African wildcats (*p* < 0.01, *d* = 0.89) and feral mixed breed (*p* = 0.02, *d* = 0.58). No significant difference in nose length was found between feral mixed breed and African wildcats (*p* = 0.84, *d* = 0.18).

The eye angles also had a main effect of group (*F*(3226) = 9.53, *p* < 0.01, *η*^2^ = 0.11). Multiple comparisons showed that owned domestic purebreds had significantly smaller eye angles than African wildcats (*p* < 0.01, *d* = 1.06), feral mixed breed (*p* < 0.01, *d* = 0.77), and owned domestic mixed breed (*p* < 0.01, *d* = 0.83). Owned domestic mixed breed did not differ significantly from African wildcats (*p* = 0.96, *d* = 0.09) and feral mixed breed (*p* = 0.98, *d* = 0.06) in terms of eye angles. Feral mixed breed and African wildcats also showed no significant difference in eye angles (*p* = 0.83, *d* = 0.15).

### 3.2. Study 2

The exact age of many cats used in Study 2 was unknown, since they were shelter cats and the age was estimated by the owner’s self-report. Using this estimated age for analysis, we found that estimated age and years with the owner were highly correlated (*r* = 0.96). Due to multicollinearity issues, we only included in our model the effect of years with the owner that we wanted to investigate in this study. Of the four variables—sex, years with the owner, sociability with people, and stranger-directed aggression—none were significantly related to nose length (Table 1), and only years with the owner were associated with eye angles (*p* = 0.04; Table 2).

### 3.3. Study 3

In terms of cuteness, human participants rated owned domestic mixed breed first (normDS = 2.28), feral mixed breed second (normDS = 1.69), owned domestic purebreds third (normDS = 1.68), and African wildcats last (normDS = 0.33) (Table 3). We also found that participants rated photoshopped pictures of African wildcats with shorter nose length as cuter than unmanipulated average faces (91 participants chose the average face while 264 chose the manipulated image: binominal test, *p* < 0.01. There was no significant difference in cuteness ratings for eye angles between the two (unmanipulated average face: 199, manipulated face: 156, *p* = 0.46). The average faces of owned domestic purebreds were found to be cuter than images of owned domestic purebreds with longer nose length (average face: 336, manipulated image: 19, *p* < 0.01). There was no significant difference in cuteness ratings for eye angles (177 participants chose the average face while 178 participants chose the manipulated image: binominal test, *p* = 1).

## 4. Discussion

In Study 1, compared with African wildcats and feral mixed breed, owned domestic mixed breed and owned domestic purebred cats were found to have neotenous facial morphology, as measured by nose length. African wildcats and feral mixed breed have essentially no contact with humans. In other words, it is suggested that interaction with humans may have caused changes in domestic cats’ facial morphology. These changes can be thought of as “transgenerational changes” (domestication) and “individual developments” (changes with years with the owner). In order to investigate which hypothesis (i.e., transgenerational changes vs. individual developments) was more plausible, in Study 2, we investigated whether interaction with humans affected the development of neotenous facial morphology in owned domestic mixed breed. We found that there was no relationship between the number of years with the owner and nose length. Furthermore, although eye angles were associated with the number of years with the owner, this is believed to be due to muscle weakening as a result of aging and not related to neotenous development. These results suggest that neotenous facial morphology in cats is a “transgenerational change” (domestication). Furthermore, Study 3 showed that this change affects cuteness ratings by humans, eliciting higher cuteness ratings for owned domestic mixed breed and owned domestic purebred cats compared with African wildcats. In summary, it is believed that cats’ facial morphology changed to display neotenous characteristics called baby schema with domestication. As baby schema elicits caregiving behavior, it may be related to the establishment of cats as companion animals.

In contrast, our results showed that the facial morphology of feral mixed breed differed from that of owned domestic mixed breed and owned domestic purebreds, but not from African wildcats. This suggests that feral mixed breed may have changed their facial morphology due to feralization. Previous studies have shown that feralization in animals does not result in morphological changes. Once domesticated, brain size reduction is considered irreversible, and in pigs, this reduction was unaffected by feralization [37]. Moreover, brain size in cats did not change from that of owned domestic cats after about 20 years of feralization [38]. However, the brains of Dingo (*Canis lupus dingo*), which are considered to be an example of feralization in canines, have been found to be larger than those of owned domestic canines of similar body size [39]. In addition, a previous study showed that there are genetic differences between stray cats in towns and feral cats in forests in Australia [40]. This supports the changes resulting from the feralization of feral mixed breed. In other words, it is possible that cats have acquired the baby schema through domestication, but as a result of feralization, their facial morphology may change again when living away from humans, as is the case with feral mixed breed.

Limitations of the present study include the following. It should be noted that we did not cover all features of the baby schema (e.g., big eyes, chubby cheeks), because cats’ faces and necks are covered in fur, making it difficult to determine their actual morphology from photographs. Thus, we concentrated on nose length. In addition, since we visually judged whether some of the photos collected from the Internet were of adult cats or not, we should analyze the information of the individuals whose age can be definitely known in the future. Study 2 was a cross-sectional survey of years with the owner; however, studying changes within the same individual cat and a longitudinal follow-up survey may be better. Due to the COVID-19 pandemic, it was not possible to conduct a household behavioral survey of the breeds’ temperaments, and only behavioral surveys were completed subjectively by the owners. In the future, we would like to consider an objective behavioral test. In Study 3, the average face of each group was created by morphing several photographs, but owned domestic purebreds’ morphology varies considerably across different breeds. Furthermore, by modelling the average face of each breed, it will be possible to determine the level of cuteness of each breed and distinguish it from the mixed breed. We used the word “kawaii” in our research, which is a Japanese word that not only means “cuteness” and evokes caretaking behaviors, but also means “looks miserable and raises sympathy” [41], which may have influenced the way the word was perceived. The word “kawaii,” when referring to cats, needs to be analyzed from a behavioral and physiological perspective.

Domestication refers to the control of behaviors and reproduction by humans, that is, changes that occur as a result of interacting with humans. However, some feral mixed breed that do not live with humans may feed on a human diet [42], and the extent to which cats interact with humans varies depending on their habitat and diet. Thus, we cannot fully eliminate the possibility that their diet influenced their facial morphology. In the future, it will be possible to clarify changes in the domestication of cats not only by distinguishing between rearing environments but also by considering and comparing individual habitats and diets. Compared with dogs and horses, which have already been domesticated, cats are said to be semi-domesticated [20]. The inclusion of diverse habitats allows the ongoing investigation of changes in the domestication of cats. Investigating these changes in feline domestication will lead to an understanding of the evolution and development of sociality in domesticated animals, including cats. Furthermore, it may contribute to the realization of a better coexistence between humans and domestic and companion animals.

## 5. Conclusions

This study showed changes in the facial morphology of cats as a result of their interaction with humans. These changes were also shown to influence humans’ cuteness ratings. Changes in the facial morphology of cats may be one of the factors that elicited caretaking behaviors in humans and established their status as companion animals. The change in facial morphology from African wildcats to domestic cats was suggested to be due to “transgenerational changes” associated with domestication, while changes in feral cats may have occurred due to feralization. Investigating changes in facial morphology in cats can lead to a better understanding of their evolution and the development of their sociality.

## Figures and Tables

**Figure 1 animals-12-03493-f001:**
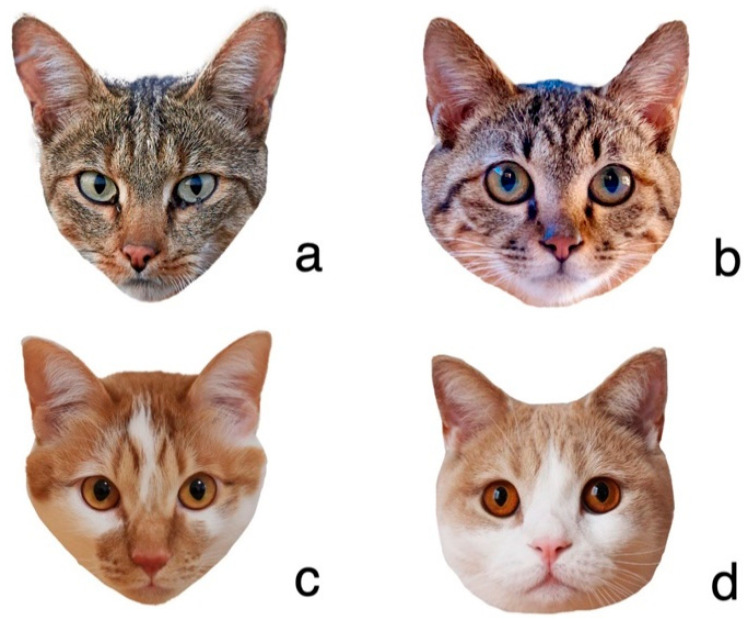
Examples of cat face photos: (**a**) African wildcat (*Felis silvestris lybica*); (**b**) feral mixed breed; (**c**) owned domestic mixed breed; (**d**) owned domestic purebred. Photo credit: (**a**) James Hager/Collection Mix: Subjects/Getty Images; (**b**) Save the Omizunagidori; (**c**) Miso; (**d**) Sakura.

**Figure 2 animals-12-03493-f002:**
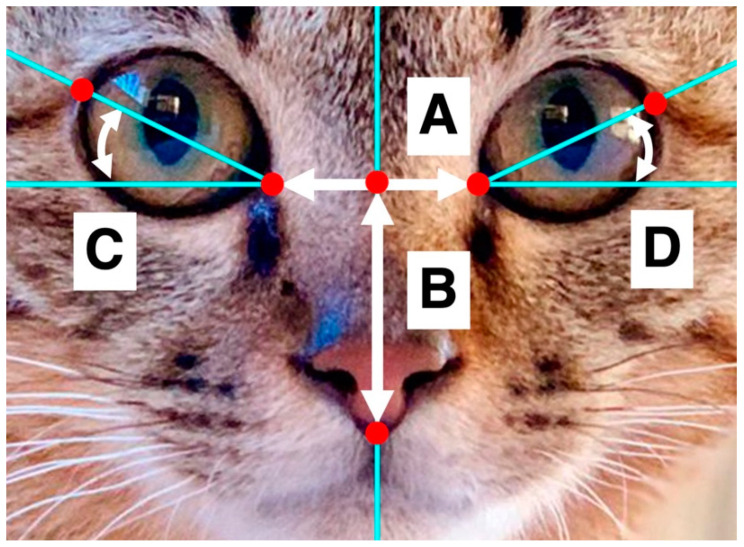
Measurement of facial morphology: A—distance between the right eye and left eye; B—length from the center line of the eye to the bottom of the nose; C—angle of the right eye; D—angle of the left eye.

**Figure 3 animals-12-03493-f003:**
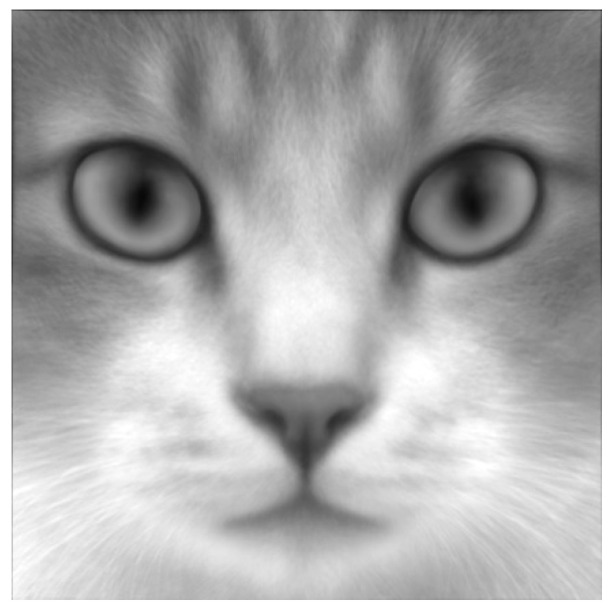
An image of the average face of owned domestic mixed breed created by morphing.

**Figure 4 animals-12-03493-f004:**
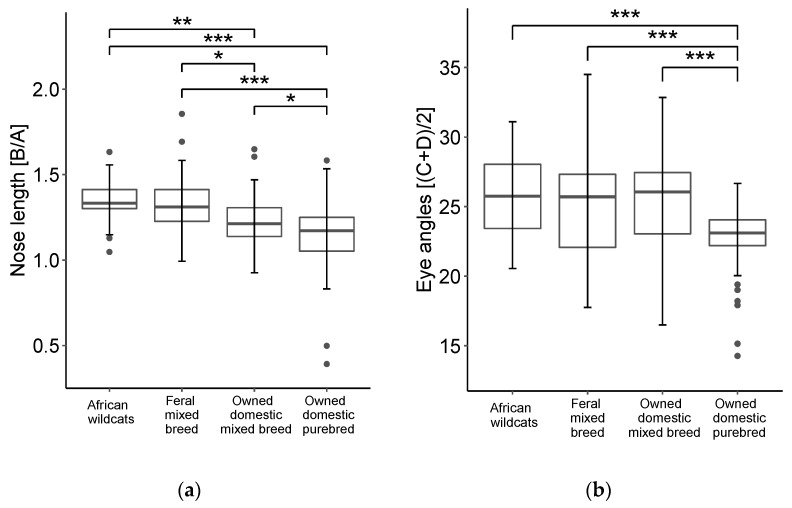
(**a**) Nose length [B/A]; (**b**) eye angles [(C + D)/2]; the bold lines represent the medians. * *p* < 0.05, ** *p* < 0.01, *** *p* < 0.001.

**Table 1 animals-12-03493-t001:** Associations between nose length and sex, years with the owner, sociability with people, and stranger-directed aggression.

Factor	Estimate	Std. Error	*t* Value	*Pr* (>*t*)
Sex	−0.01	0.01	−0.42	0.67
Years with the owner	−0.00	0.00	−0.50	0.62
Sociability with people	−0.00	0.01	−0.37	0.71
Stranger-directed aggression	−0.00	0.01	−0.14	0.89

**Table 2 animals-12-03493-t002:** Associations between eye angles and sex, years with the owner, sociability with people, and stranger-directed aggression. * *p* < 0.05.

Factor	Estimate	Std. Error	*t* Value	*Pr* (>*t*)
Sex	−0.48	0.32	−1.49	0.14
Years with the owner	−0.08	0.04	−2.06	0.04 *
Sociability with people	−0.24	0.13	−1.83	0.06
Stranger-directed aggression	−0.03	0.16	−0.19	0.85

**Table 3 animals-12-03493-t003:** Comparison of the four nonmanipulated groups calculated from David’s score.

	African Wildcats	Feral Mixed Breed	Owned Domestic Mixed Breed	Owned Domestic Purebred	David‘s Score	normDS	Rank
African wildcats	-	48	25	44	−4.6	0.33	4
Feral mixed breed	307	-	110	183	0.78	1.69	2
Owned domestic mixed breed	330	245	-	236	3.12	2.28	1
Owned domestic purebreds	311	172	119	-	0.75	1.69	3

## Data Availability

The data presented in this study are available in Appendix A.

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
