# Peer review of "Changes in Cat Facial Morphology Are Related to Interaction with Humans"

_animals, 2022, doi:10.3390/ani12243493_

Round 1

Reviewer 1 Report

This paper investigated the effect of human interaction on neotenous face morphology in cats by comparing it among cats (feral mongrels, owned domestic mongrels, and owned domestic purebreds) and African wildcats. In study 1, they measured the relative nose position and eye angle of these four types and found that owned cats are different from African wildcats and feral mongrels, but the latter two are not different between each other. In Study 2, they investigated how years of breeding affects these face morphologies in owned cats but did not find a significant effect. In Study 3, by using manipulated stimuli, the authors tested human cuteness rating for different type of cat faces and found that nose length affects it. They conclude that domestication and feralization affect neotenous face morphology (nose length) in cats and it affects human preference.

Overall, the findings in the paper are interesting. The effect of human interaction on face morphology in companion animals attracts attention these days. It is nice that the authors compared cats with three different levels of human interactions to differentiate the morphological changes caused by transgenerational factors and individual developments. The sample size is seemingly big enough. However, there are several things that need to be clarified.

My first main concern is that the authors assume two-step changes (domestication and feralization) in cat face morphology, but it is not clear to me how the authors can deny other possibilities. Given that only owned domestic cats are different from African wild cats and feral cats, it is possible that this semi-domestication and feralization have not changed cats’ face morphology, but only recent rearing environments (e.g., high nutrias food) changed it in owned domestic cats. I think this explanation is simpler than assuming two step changes.

This first concern could have been tested in Study 2, where the effect of breeding years (the length of owned period) on cats’ face morphology was investigated. However, my second concern is that the age of cats is not included in the analysis, which makes it difficult to tell apart the effect of breeding years from the effect of age.

Besides these points, some parts are not very clear. The followings are my comments.

General comments

l  How genetically different feral cats are from owned cats? The explanation specifically about the analyzed feral cats (e.g., approximately when they are feralized) will also be informative.

l  What is the intention to compare owned domestic mongrels and owned domestic purebreds? How the authors explain the reason why nose length and eye angles differ between owned domestic mongrels vs. owned domestic purebred?

l  Age of analyzed cats is important. How did the authors know the age (or at least that the cat is mature) when they took images from Internet?

Specific comments

l  L49. “Facial morphology is important for a companion animal in eliciting caretaking behaviors from humans.”

This paragraph is not easy to follow because the arguments are going back and forth. Discussing evidence of human preference for neonatal features in companion animal faces and then discussing opposite evidence will make the paragraph clearer.

You may want to mention the examples that domestication changed face morphology and affect human caretaking behaviour in another companion animal, dogs by (Kaminski et al., 2019; Waller et al., 2013).

-Kaminski, J., Waller, B. M., Diogo, R., Hartstone-Rose, A., & Burrows, A. M. (2019). Evolution of facial muscle anatomy in dogs. Proceedings of the National Academy of Sciences, 116(29), 14677–14681. https://doi.org/10.1073/pnas.1820653116

-Waller, B. M., Peirce, K., Caeiro, C. C., Scheider, L., Burrows, A. M., McCune, S., & Kaminski, J. (2013). Paedomorphic facial expressions give dogs a selective advantage. PLoS ONE, 8(12), e82686. https://doi.org/10.1371/journal.pone.0082686

l  L78. “If domestication indeed altered brain and skull size, as well as neotenous facial morphology, what process brought about this change?
This question should be rephrased because individual development is not specifically related to domestication. Maybe it’s better to say:
“If human interaction indeed influences neotenous facial morphology as well as brain and skull size,
what process brought about this change?

l  L81. “their interaction with humans”
It could be misleading because one may think that (daily) interaction with humans is more related to “individual developments” instead of “transgenerational changes”.

l  L135. (and other places) “Nose length”
Is it the same as “the relative position of their eyes and nose” (e.g., L288.and L355)?
 I think the relative position of their eyes and nose is more accurate words.

l  The results in Study 1
Nose length is commonly mentioned as one of the characteristics of baby schema but eye angle is not (and indeed it does not influence on cuteness rating in Study 3). How do the authors interpret the effect of human interaction on eye angle?

l  The analysis in Study2
Why didn’t the author include the age of the cat into the model? It is impossible to differentiate the effect of years of breeding
from that of age as authors mentioned in discussions.

l  Stimuli in Study 3,
The description of stimuli is not clear. For example, “the average face photographs of the four groups were combined into a single photograph” is not clear to me. Showing example of the stimuli is also helpful.

l  The paragraphs about the stimuli and the combination (L191-220)
These look much more complicated than what it is and it is not easy to understand the reason for each comparison. I strongly recommend explaining stimuli and the comparison among (1)-(4) first, then (1) vs. (5)(6), and then (4) vs. (7)(8), something like:

“First, we made average faces of four types of cats, African wildcats, Feral mongrels…. We compared all six possible combinations among four average faces. Then, in order to investigate the effects of nose length and eye angles, we created African cat face with either shorter nose length or with smaller eye angle, and compared them with the original average face of African cats…”

l  Were the average faces of Owned domestic purebreds were made from different breeds? If so, what is the difference between the average face of owned domestic mongrels and owned domestic purebreds? How do the authors explain the reason why mongrel was rated cuter than purebred?

l  L300. “These results suggest that neotenous facial morphology in cats is a “transgenerational change” (domestication)”

I’m not sure if the authors can call it as domestication effect because there is no difference between African wildcats and feral mongrels in Study 1. As they mentioned, it may be possible that feral mongrels experienced a two-step change by domestication and feralization. However, it is simpler to assume that only owned cats experienced morphological change recently by environmental factors (e.g., high nutrias food, indoor environment) instead of domestication or feralization.

L343. “Further, we will be able to clarify the human cuteness rating of owned domestic purebreds by classifying the characteristics of facial morphology for each breed and creating a model of the average face.”
This sentence is not very clear to me.

That is all.

Reviewer 2 Report

General comment:

The term “mongrel” is used throughout the paper, but this term typically refers to dogs, not cats. I suggest using “mixed breed” instead. Also, it would be useful to explicitly define in the Introduction the four groups you are comparing. For example, in the U.S., some feral cats are fed by humans whereas it sounds like your feral cats were not (line 122).

I suggest adding feralization and transgenerational change as keywords, and possibly domestication syndrome.

Lines 97-100 repeat information already discussed in the two paragraphs above – delete.

Lines 103-108: This information on measurements should only be in the Materials and Methods section (lines 133-142) rather than in both places.

Line 110 and 163-164: In Study 2, two phrases are used to describe the same variable, “years of living with humans” and “years of breeding.” Please clarify these phrases here and throughout the paper, so readers will understand this variable. Are you referring to age of the cats?

Line 126: How could you verify that photographs from the Internet were of feral cats, and not, for example, owned cats outside? Was this based on location (e.g., the photos were from the Japanese islands sampled)?

Lines 163-167: Did you have survey participants complete all sections of Fe-BARQ or just those related to sociability with people and stranger-directed aggression? Also, if possible, it would be helpful to provide the exact questionnaire used as Supplementary Material. I imagine that there was a section on demographic characteristics of the cats yet only the sections of Fe-BARQ are provided in Table S-2.

Line 176: Delete “on humans.”

Results section: Most journals require the use of leading zeros when reporting p values (e.g., p < 0.05). There are many places where the p values are reported as p < .00. I know that some statistical packages report p values of 0.000, but I have never seen p values reported as such in papers. Would p < 0.001 be more appropriate?

Lines 287- 289: Based on the introduction (lines 106-108), I thought eye angles were not part of the baby schema yet here they are discussed with reference to neotenous facial morphology. Also, wouldn’t “length of nose” be more accurate in this sentence than “relative position of their eyes and nose”?

Lines 307-312: If you clearly define the four groups under study early in the paper, as suggested above, then this paragraph could be shortened considerably because it describes all the groups again.

Lines 323-330: This paragraph seems to repeat some of the information in lines 307-312. I suggest combining these paragraphs and removing the repetitive information. The paragraph on feralization in other species (lines 313-322) should follow discussion of your specific results (right now it is between two paragraphs on your findings).

Line 336: Again, I would say “length of nose” here.

I enjoyed reading this paper and hope the authors find my comments helpful.

Round 2

Reviewer 2 Report

I reviewed an earlier version of this manuscript and find the revised manuscript much improved. I have just one remaining concern and two minor comments. In my initial review, I asked for clarification regarding the phrases “years of living with humans” and “years of breeding” and in your cover letter you explained that originally you used the phrases interchangeably, but in the revision settled on “years of breeding.” My concern about the phrase  “years of breeding” is that it sound like the variable refers to “years cats were producing litters,” when many of the owned cats were likely spayed/neutered. This variable is important in Study 2, so its meaning still requires clarification. I checked the Duffy et al. (2017) reference for Fe-BARQ, and they used “Age acquired by the owner” and gave several ranges as options (e.g., from < 2 months to > 10 years). Do you mean Age acquired by owner? Or possibly “Number of years the cat lived with the owner at the time of the survey” - this could be calculated from age acquired and date of survey. Note that this will require changes in the text and Tables 1 and 2.

Line 216: Editing is needed for the words “for nose length of their eyes and nose.”

Figure 4b: Should the label on the Y axis be “Eye angles [C+D/2]” rather than “Eye angles [CD]?” If so, please change in the caption for Figure 4b as well.
